# In Vitro Newly Isolated Environmental Phage Activity against Biofilms Preformed by *Pseudomonas aeruginosa* from Patients with Cystic Fibrosis

**DOI:** 10.3390/microorganisms9030478

**Published:** 2021-02-25

**Authors:** Ersilia Vita Fiscarelli, Martina Rossitto, Paola Rosati, Nour Essa, Valentina Crocetta, Andrea Di Giulio, Veronica Lupetti, Giovanni Di Bonaventura, Arianna Pompilio

**Affiliations:** 1Cystic Fibrosis Diagnostics, Microbiology and Immunology Diagnostics, Bambino Gesù Children’s Hospital (OBG), 00165 Rome, Italy; evita.fiscarelli@opbg.net (E.V.F.); martina.rossitto@opbg.net (M.R.); nour.essa@opbg.net (N.E.); 2Clinical Pathways and Epidemiology, Bambino Gesù Children’s Hospital OBG, 00165 Rome, Italy; 3Department of Medical, Oral and Biotechnological Sciences, Center for Advanced Studies and Technology (CAST), “Gabriele d’Annunzio” University of Chieti-Pescara, 66100 Chieti, Italy; valentina.crocetta@unich.it (V.C.); veronica.lupetti@studenti.unich.it (V.L.); gdibonaventura@unich.it (G.D.B.); arianna.pompilio@unich.it (A.P.); 4Department of Science, Interdepartmental Laboratory of Electron Microscopy, L.I.M.E., Roma Tre University, 00146 Rome, Italy; andrea.digiulio@uniroma3.it

**Keywords:** newly isolated environmental phages, bacteriophage in vitro activity, phage efficacy, *Pseudomonas aeruginosa*, biofilm, planktonic cells, antibiotic resistance, chronic lung infections, cystic fibrosis

## Abstract

As disease worsens in patients with cystic fibrosis (CF), *Pseudomonas aeruginosa* (PA) colonizes the lungs, causing pulmonary failure and mortality. Progressively, PA forms typical biofilms, and antibiotic treatments determine multidrug-resistant (MDR) PA strains. To advance new therapies against MDR PA, research has reappraised bacteriophages (phages), viruses naturally infecting bacteria. Because few in vitro studies have tested phages on CF PA biofilms, general reliability remains unclear. This study aimed to test in vitro newly isolated environmental phage activity against PA isolates from patients with CF at Bambino Gesù Children’s Hospital (OBG), Rome, Italy. After testing in vitro phage activities, we combined phages with amikacin, meropenem, and tobramycin against CF PA pre-formed biofilms. We also investigated new emerging morphotypes and bacterial regrowth. We obtained 22 newly isolated phages from various environments, including OBG. In about 94% of 32 CF PA isolates tested, these phages showed in vitro PA lysis. Despite poor efficacy against chronic CF PA, five selected-lytic-phages (Φ4_ZP1, Φ9_ZP2, Φ14_OBG, Φ17_OBG, and Φ19_OBG) showed wide host activity. The Φ4_ZP1-meropenem and Φ14_OBG-tobramycin combinations significantly reduced CF PA biofilms (*p* < 0.001). To advance potential combined phage-antibiotic therapy, we envisage further in vitro test combinations with newly isolated phages, including those from hospital environments, against CF PA biofilms from early and chronic infections.

## 1. Introduction

Cystic fibrosis (CF), a common genetic disease among Caucasian populations, has a typically poor prognosis related to recurrent bacterial lung infections. The leading cause of respiratory failure and mortality is the Gram-negative opportunistic environmental pathogen *Pseudomonas aeruginosa* (PA), a bacterium that colonizes 30% of children and up to 80% of adults with CF [1,2,3]. In these patients, PA lung infections owe their life-threatening severity to their ability to switch from the wild-type to the mucoid phenotype, namely over-producing alginate, and forming biofilm [4]. This biofilm lifestyle prevents antibiotics from eradicating PA, especially multidrug-resistant (MDR) PA [5]. To overcome the current lack of research on new antibiotics against MDR PA, Western countries have reappraised research on bacteriophages (phages), viruses that naturally infect bacteria [6,7,8]. Since the past century, these viruses, commonly found in the environment (sewage and dirty water), have attracted interest as a promising therapeutic alternative to antibiotics [8]. In the past years, phages have shown proven efficiency also against biofilms formed by non-CF and CF PA isolates [8,9]. Promising results also come from studies on phage therapy, alone or combined with antibiotics in treating CF PA infections, in in vitro biofilm models, and lung infected animal models [8,9,10].

Despite these findings, research has focused almost entirely on phages infecting laboratory PA or non-CF PA isolates from acute or chronic infections, but relatively few in vitro studies aimed to test phages on clinical CF PA biofilms, thus reducing general reliability [10].

To advance information on phage therapy in clinical biofilm producer CF PA isolates in in vitro experiments, we deemed it important to test newly isolated environmental phages on several CF PA biofilms, alone or combined with selected antibiotics, thus providing new information on potential phage-based clinical treatment.

This study aimed to test in vitro newly isolated environmental lytic bacteriophages against several PA isolates from sputum in various lung infection stages in patients with CF, treated at Bambino Gesù Children’s Hospital (OBG) in Rome, Italy. To do so, we first tested in vitro the lytic phage activity alone against CF PA isolates. Then, we tested newly isolated selected phages, according to their activity, alone and combined with three anti-pseudomonal antibiotics—amikacin (AMK), meropenem (MPM), and tobramycin (TOB)—against several CF PA biofilms. We also investigated, in in vitro CF PA phage-infected biofilm supernatants, whether new PA colony morphotypes appear, and in CF PA biofilms and planktonic cell cultures whether bacterial regrowth develops.

## 2. Materials and Methods

### 2.1. Laboratory PA, CF PA Isolates and Culture Conditions

Overall, we tested two laboratory PA (PAO1 and PA14) and six clinical PA (MA1-MA6) isolates, chosen as indicator bacteria (IB) for their sensitivity to phage infection, thus allowing phage replication and isolation. We also tested another 32 CF PA isolates collected from patients with CF treated at OBG, recorded as Pa_Ph1–Pa_Ph32. We defined CF PA isolates according to three lung infection stages as first infection (PA isolated right after the acquisition, namely from Pa_Ph1 to Pa_Ph9), early infection (PA isolated one year after the first infection, namely from Pa_Ph10 to Pa_Ph15), and chronic infection caused by non-mucoid or mucoid PA isolates (PA isolated for at least five years after first infection, namely non-mucoid isolates from Pa_Ph16 to Pa_Ph25, and mucoid isolates from Pa_Ph26 to Pa_Ph32). We cultured all the CF PA isolates overnight at 37 °C in trypticase soy broth (TSB, bioMérieux, Marcy-Letolle, France). We then centrifuged overnight cultures at 2500× *g* for 10 min (Megafuge 1.0, Heraeus Instrument, Hanau, Germany), and resuspended them in sulfate magnesium (SM) buffer (100 mM NaCl, 8 mM MgSO_4_, 50 mM Tris-HCl, pH 7.5).

### 2.2. Phage Isolation and Purification

After collecting 12 sewage samples from different sources, nine from farm-house sewage and a river in Viterbo, Italy, two from municipal sewers in Rome, and one from sewers at OBG, Rome, Italy, phages were isolated and purified by the double agar layer method [11]. Each sample was centrifuged at 12,000× *g* for 10 min, and then filtered supernatants with a 0.45 µm pore size filter (Minisart filter; Sartorius, Göttingen, Germany). A hundred μL of sewage filtrate was added to 200 μL of each IB, mixed with 3.5 mL melted trypticase soy top-agar, and poured on the surface of trypticase soy agar (TSA) plates (Becton Dickinson, Franklin Lakes, NJ, USA). After overnight incubation at 37°C, these TSA plates were checked to identify lytic plaques and classify them according to their morphological characteristics including size, aspect (clear or turbid), and halo absence or presence [12]. Single plaques were then selected and resuspended in SM buffer. To purify lytic phages, the whole procedure was repeated at least three times, thus producing a high titer lysate plaque-forming unit (PFU/mL), as verified by the spot-test [13]. After processing, purified phages were stored at −80 °C in the laboratory phage bank at OBG.

### 2.3. Morphological Phage Characterization

Before characterizing phages, lytic phages were selected. Their morphology was evaluated by electron microscopy, to assign them an order and family in accordance with the International Committee on Taxonomy of Viruses (ICTV) criteria (https://talk.ictvonline.org/ictv-reports/ictv_9th_report/dsdna-viruses-2011/w/dsdna_viruses/69/podoviridae, accessed on 15 January 2021). To do so, two different electron microscopy techniques were used, scanning electron microscopy (SEM) and transmission electron microscopy (TEM). Two completely different phage sample preparations were obtained to render comparable and complementary images on phage morphology and dimensions. These experimental techniques minimized possible artefacts due to differences in sample preparation for the two techniques, and microscope operating.

SEM phage characterization: After incubating 100 μL of phage lysate and 200 μL of PAO1, Karnovsky fixative (Electron Microscopy Sciences, Hatfield, PA, USA) was added, removed by centrifugation, and then samples were dehydrated by serial passages up to 100% EtOH (Kaltek, Padua, Italy). Pellet samples were dried with hexamethyldisilazane (Sigma-Aldrich, Darmstadt, Germany), mounted on aluminum stubs (Ted Pella Inc., Redding, CA, USA), gold-coated in an Emitech K550 unit, and finally phages were examined under the electron column at the operating voltage of 2 and 5 kV (DualBeam FIB/SEM Helios Nanolab, FEI Company, Eindhoven, The Netherlands) with the secondary electron detector for obtaining high-resolution SEM phage images (L.I.M.E., Roma Tre University, Rome, Italy).

TEM phage characterization: A drop of high-titer phage suspension (10^9^–10^10^ PFU/mL) was adsorbed onto a formvar carbon film copper grid (Sigma-Aldrich, Darmstadt, Germany) (300 mesh), fixed with 2% glutaraldehyde (Electron Microscopy Sciences, Hatfield, PA, USA) in PBS (Sigma-Aldrich, Darmstadt, Germany), and negatively stained with 2% phosphotungstic acid in H_2_O (pH 7) (Bio-Optica, Milan, Italy). Phage preparations were then observed at an operating voltage of 80 kV under a Zeiss EM 900 electron microscope (Carl-Zeiss, Oberkochen, Germany). To avoid possible phage damage provoked by the high voltage used, the grids were also observed with the electron beam operated at 30 kV and a 0.17 nA current under the Scanning Transmission Electron Microscopy (STEM) detector (Dualbeam FIB/SEM Helios NanoLab 600, FEI Company, Hillsboro, OR, USA). Finally, TEM and STEM techniques were used to compare the same phage sample images.

To measure phage heads and tails, the Helios Nanolab 600 proprietary software “xT microscope Control” (FEI Company, Hillsboro, OR, USA) was used on SEM, TEM, and STEM phage images acquired at 2048 × 1768 pixels.

### 2.4. Functional Phage Characterization

Phage latency and burst size were examined in a modified single-step growth experiment [13]. Using phages at a multiplicity of infection (MOI) of 0.001, the laboratory PAO1 was challenged in a mid-exponential growth phase, thus avoiding multiple phage-PAO1-adsorption, and distinguishing the various phages. For phage titration, we immediately removed a 100 μL aliquot from this phage-PAO1 preparation, and let the remaining phages adsorb on PAO1 as a phage-PAO1 co-culture for 15 min at 37 °C. This co-culture was diluted with TSB and incubated at 37 °C for 1 h. During incubation, every five min samples were taken for phage titration using phage spot-tests and plaque assays. For functional phage characterization, after incubating plates overnight at 37 °C, PFU were enumerated by defining phage titers during the latent period and at a plateau level, and burst sizes were determined as PFU/cell [13]. For each phage burst size ratio, the following formula was used [13]:(1)$$\mathrm{Phage}\;\mathrm{burst}\;\mathrm{size}=\frac{\mathrm{phage}\;\mathrm{titer}\;\mathrm{when}\;\mathrm{the}\;\sin\mathrm{gle}\;\mathrm{step}\;\mathrm{growth}\;\mathrm{curve}\;\mathrm{ended}\;–\;\mathrm{phage}\;\mathrm{titer}\;\mathrm{when}\;\mathrm{PFUs}\;\mathrm{started}\;\mathrm{to}\;\mathrm{increase}}{\mathrm{phage}\;\mathrm{titer}\;\mathrm{when}\;\mathrm{PFUs}\;\mathrm{started}\;\mathrm{to}\;\mathrm{increase}}$$

### 2.5. Cross-Activity Phage Evaluation

Each phage was tested for cross-activity toward laboratory PAO1 and PA14 strains, and all CF PA isolates. We used each PA isolate to inoculate a TSA plate according to the double layer agar method, as described in the foregoing. Once the top agar hardened, we spotted 10 μL of purified phage suspensions on it, thus scoring the lytic phage spectra activity, according to the phage characteristics, namely 0 no clearing, +1 few individual plaques, +2 substantial turbidity throughout a cleared zone, +3 clearing throughout a faintly hazy background, and + 4 complete clearing [14].

### 2.6. Phage Activity against PA Biofilms

To test phage activity, after incubation for 24 h at 37 °C with TSB, one IB (MA4) was checked and renamed Pa_Ph33, and all the 32 CF PA isolates were checked for their ability to produce biofilm in 96-well polystyrene microplates. After crystal violet staining, PA biofilm biomasses in the microplates were assessed spectrophotometrically by measuring optical density at 492 nm (OD_492_) [15]. According to lytic phage spectra detected from the previous cross-activity phage evaluation, to assess the anti-PA biofilm phage effect, five lytic phages were selected, namely Φ4_ZP1, Φ9_ZP2, Φ14_OBG, Φ17_OBG, and Φ19_OBG. To do so, each phage was tested for its ability to reduce biofilm biomass in the CF PA isolates, and PAO1 by choosing phages according to their shown sensitivity in spot-tests. Phage Φ4_ZP1 was tested against 21 PA, Φ9_ZP2 against 17 PA, Φ14_OBG against 22 PA, Φ17_OBG against 31 PA, and Φ19_OBG against 24 PA isolates. A 24-h-old PA biofilm was used to test phage exposure at MOI 1 (1–2 × 10^7^) and 100 (1–2 × 10^9^ PFU/mL), by replacing spent TSB with a phage suspension in TSB + SM buffer (1:1 *v/v*). To assess a negative control, phage-untreated PA biofilm samples were exposed to TSB + SM buffer alone. All the PA biofilm samples (phage-infected PA, and negative controls) were incubated at 37 °C for 4 to 24 h, and then PA biofilm dispersion was evaluated with a spectrophotometer in samples stained with crystal violet.

### 2.7. Phage and Antibiotic Combined Treatment Effect on PA Biofilms

Experiments were done on biofilms formed by nine CF PA isolates (Pa_Ph3, Pa_Ph7, Pa_Ph8, Pa_Ph9, Pa_Ph10, Pa_Ph12, Pa_Ph18, Pa_Ph20, and Pa_Ph21), because they showed the highest sensitivity to phage exposure, and these biofilms were exposed to phages for 4 h at an MOI chosen according to their best activity against PA biofilm. Each phage-CF PA biofilm was tested with AMK, MPM, and TOB antibiotics (all from Sigma-Aldrich, Milan, Italy) at their minimal inhibitory concentration (MIC) value, as assessed by broth micro-dilution technique according to the European Committee on Antimicrobial Susceptibility Testing (EUCAST) guidelines (Version 11.0, 2021, https://www.eucast.org/clinical_breakpoints/, accessed on 15 January 2021). Negative controls were untreated biofilms exposed to medium alone. A spectrophotometer was then used to measure crystal violet stained PA biofilm dispersions after phage exposures, antibiotics, and their combinations [15].

### 2.8. Supernatant PA Phenotypes and Antibiotic Sensitivity Tests

After exposing PA biofilms to phages (at MOI 1 and 100) for 24 h, and before adding crystal violet, 100 μL of the supernatant were collected from one well for each non-mucoid PA-phage combination, and plated each onto MacConkey agar plates (Oxoid, Milan, Italy). After incubation at 37 °C for 48 h, cultures were checked for the presence of PA mucoid phenotype and, if present, PA mucoid colonies were sub-cultured three times onto MacConkey agar plates (37 °C, 24 h). The stable PA phenotypes were then stored at −80 °C for further investigations. To do so, the Kirby-Bauer disk diffusion technique was used to test ancestral (wild type, WT) CF PA isolates, and the relative mucoid phenotypes detected after phage exposure with several antibiotics: AMK, aztreonam (ATM), cefepime (FEP), ceftazidime (CAZ), ciprofloxacin (CIP), imipenem (IPM), levofloxacin (LVX), MPM, piperacillin-tazobactam (TZP), and TOB. Results were interpreted and reported according to the EUCAST criteria (Version 11.0, 2021, https://www.eucast.org/clinical_breakpoints/, accessed on 15 January 2021).

### 2.9. Phage-CF PA Planktonic Cell Infection Kinetics

To assess phage-CF PA planktonic cell infection kinetics, Pa_Ph4 and Pa_Ph6 isolates cultured overnight were corrected at OD_550_ 1.0 (1–4 × 10^8^ colony forming units, CFU/mL), and then diluted 1:10 with TSB-SM (1:1 *v/v*). After 2 h, phages were added at MOI 10, and samples incubated at 37 °C for a further 22 h. To test negative control samples, PA planktonic cells were exposed to TSB-SM alone. During the first 6 h post-infection, viable PA cells were counted hourly and 22 h after the phage infection. Finally, when phage infection began and ended, the spot assay was used to count PFUs [12].

### 2.10. Statistical Analysis

Differences between frequencies were assessed by the χ^2^ test using GraphPad Prism 5.0 (GraphPad Software, Inc., San Diego, CA, USA). *p* values less than 0.05 were considered statistically significant.

## 3. Results

### 3.1. Phage Isolation and Purification

The laboratory PAO1 and CF MA4 IB isolates yielded 22 previously undescribed new phages. For each phage, we assessed plaque morphology (size, aspect, presence or absence of halo), title, and cross activity versus other IB (Appendix A). No phage was active toward MA1, MA2, MA3, and MA5 CF isolates. Conversely, when the purified phage stocks were tested against the whole IB isolate collection, some phages showed cross-activity. Specifically, all phages isolated on MA4 were able to infect PAO1, MA1, and MA3 CF isolates. In addition, phages Φ4_ZP1 and Φ 9_ZP2 infected two PA isolates, MA5 and MA6, from adult patients with CF. Conversely, all seven phages isolated on PAO1 strain showed a wider host range. Hence, when the PAO1 strain was used for the three infection cycles needed for isolating single phages, each of the 7-phage stock infected various CF PA isolates (MA1, MA2, MA3, MA4, and MA5) (Appendix A).

### 3.2. Morphological Phage Characterization

Phage morphology for the phages representing various host ranges (Φ4_ZP1, Φ9_ZP2, Φ14_OBG, Φ17_OBG, and Φ19_OBG) was characterized in detail (Table 1). By TEM and STEM phage characterization, along with SEM observation during the PAO1 infection, all phages with an icosahedral head and a tail were identified. According to the ICTV criteria, we identified phages as belonging to the *Caudovirales* order, and the *Podoviridae* family, characterized by a short and non-contractile tail (Figure 1).

### 3.3. Functional Phage Characterization

In single-step growth experiments designed to assess phage growth characteristics (Table 1), phage Φ4_ZP1 exhibited the shortest latent period (10 min) and phage infection cycle (20 min). Conversely, the phage Φ17_OBG exhibited the longest infection cycle 50 min, and Φ19_OBG the longest latent period. Phage Φ17_OBG was the most prolific (burst size 34 ± 18 PFU/cell), and phage Φ19_OBG the least prolific (burst size 4 ± 0.5 PFU/cell) (Table 1).

### 3.4. Cross Activity Phage Evaluation on Clinical CF PA

For the 22 newly isolated phages identified, phage host range and lytic ability against 32 CF PA isolated from the three lung infection stages, previously defined by a spot-test method, ranged from 34.4% (11 PA isolates lysed by phage Φ3_ZP1) to 93.7% (30 PA isolates lysed by phage Φ17_OBG) (Figure 2). All the phages isolated on PAO1 (Φ16-22_OBG) showed a significantly wider host range than did phages isolated on MA4 (Φ1-2_DL1, Φ3-5_ZP1, Φ6-10_ZP2, Φ11-12_ZP3, Φ13-15_OBG). The seven phages isolated on PAO1 showed higher lytic activity than those isolated on MA4 (range: 69.7%–90.9% vs. 36%–60.6%; mean: 80.9% vs. 51.5%; *p* < 0.05). Specifically, despite being insensitive to phages isolated on MA4, 12 CF PA isolates were efficiently killed by phages isolated on PAO1. Phages isolated on PAO1 killed most chronic mucoid PA isolates (Figure 2).

Stratifying phage lytic abilities according to the infection stages and CF PA phenotypes, showed that all 22 phages tested lysed four of the nine CF PA isolates (44.4%) from first infections (Figure 2). Mucoid CF PA isolates (from early and chronic infections), and chronic non-mucoid CF PA isolates showed a similar sensitivity pattern to phage predation. Mucoid and chronic non-mucoid CF PA isolates showed lower phage sensitivity than PA isolated during first or early infections (49% vs. 72%) (Figure 2).

### 3.5. Phage Activity against Preformed PA Biofilms

When 32 CF PA isolates were screened for their ability to develop biofilm in 96-well polystyrene microplates, all did so despite various biomass levels. Testing combinations showing any phage activity by the spot-test method, after 4 h phage exposures, disclosed that the PA biofilm biomass diminished significantly in 42% (42 out of 100) at MOI 1, and in 53.9% (55 out of 102) at MOI 100 (Table 2; Appendix A).

Although in vitro phage activity in reducing-PA biofilm biomasses was unrelated to the phage MOI used after 4 h exposures, the two phages Φ9_ZP2 and Φ19_OBG at MOI 100 reduced PA biofilms most effectively (Φ9_ZP2 reduced PA biomasses in 10 out of 15 phage-PA isolate combinations, 66.6%; and Φ19_OBG in 14 out of 24 combinations, 58.3%). Although in nearly all the five newly isolated lytic phages tested, the MOI used left the ability to reduce PA biofilm biomasses unaffected, the phage Φ9_ZP2 showed a statistically significant difference in reducing PA biofilm biomasses at MOI 1 versus that at MOI 100 after 4 h exposures (26.6% and 66.6%; *p* < 0.001) (Table 2; Appendix A).

Prolonging CF PA biofilm-phage exposure times from 4 to 24 h, left PA biofilm biomass reductions significantly unchanged (at MOI 1 from 42% to 36.5%, and at MOI 100 from 53.9% to 43.4%) (Table 2; Appendix A). Even though overall CF PA biofilm biomasses exposed to phages at MOI 1 for 24 h showed similar biofilm biomass reductions, the Φ14_OBG tested at MOI 100 at 24 h reduced CF PA biomasses more effectively than Φ4_ZP1 and Φ17_OBG (Φ14_OBG 68.1% vs. Φ4_ZP1 33.3% *p* < 0.05; and Φ14_OBG 68.1% vs. Φ17_OBG 32.2% *p* < 0.01) (Table 2).

Independently from the phage and the MOI used, CF PA biofilm biomass increased after 4 and 24 h of phage-exposures, mainly in Pa_Ph6 and Pa_Ph22 isolates.

### 3.6. Phage and Antibiotic Combined Treatment Effect on PA Biofilms

According to the observed phage activity against PA biofilms, when tested in vitro the five phages (Φ4_ZP1, Φ9_ZP2, Φ14_OBG, Φ17_OBG, and Φ19_OBG) alone, and AMK, MPM, and TOB antibiotics at sub-MIC concentrations alone against 24-h-old CF PA biofilms formed by nine PA isolates (Pa_Ph3, Pa_Ph7, Pa_Ph8, Pa_Ph9, Pa_Ph10, Pa_Ph12, Pa_Ph18, Pa_Ph20, and Pa_Ph21), PA biofilm biomasses increased with phages alone by about 5.5%, and 14.8% when exposed to antibiotics alone. Specifically, Φ4_ZP1 alone and AMK alone against Pa_Ph21 biofilm, increased biofilm biomasses to a similar significant extent (Φ4_ZP1-Pa_Ph21 143.4%; and AMK-Pa_Ph21 179.4 %) (Figure 3). Similarly, when AMK alone against Pa_Ph18 biofilm was compared versus the negative control in TSB, the PA biofilm markedly increased (572.9 %) (Figure 3).

When all the PA biofilms were exposed to MPM alone, no PA biofilm biomasses increased (Figure 4).

Results from crystal violet assays also showed in Pa_Ph21 biofilm biomasses exposed to TOB alone a significant increase versus phage-untreated controls in TSB (291.8 %) (Figure 5). In the Pa_Ph21 biofilm, pre-treated with Φ4_ZP1 and then exposed to TOB, despite the initial biofilm increase, the biofilm returned to a level comparable to the matched negative control in TSB (Figure 4).

Overall, when spectrophotometer readings comparing the PA biofilms treated with the phage-antibiotic combinations after 4 h with those treated with the five phages alone, PA biofilm biomasses diminished significantly in about 50% (Figure 3, Figure 4 and Figure 5).

Specifically, seven of the 18 phage-AMK combinations tested reduced PA biofilms significantly more than the five phages alone (38.9% total PA biofilm reductions) (Figure 3). Although no significant difference emerged between the phage-AMK combinations compared with phages alone or AMK alone, phage Φ17_OBG combined with AMK reduced PA biofilm biomasses in 40% of the combinations (Figure 3).

In 12 of the 18 phage combinations tested, phage-MPM combinations reduced PA biofilm biomasses significantly more than the five phages alone (66.6% total PA biofilm reductions) (Figure 4). The most effective phage-MPM combination was Φ4_ZP1, which reduced PA biofilm biomass in 50% of 4 phage-MPM combinations tested compared to MPM alone (Figure 4). Overall, MPM alone was more effective in reducing PA biofilm biomasses than phages alone in 72.2% combinations tested, whereas phages alone were more effective than MPM alone in reducing the PA biofilm biomasses in 11.1% of the PA biofilms treated (*p* < 0.001) (Figure 4).

In nine of the 18 phage combinations tested, phage-TOB combinations reduced PA biofilms significantly more efficiently than the five phages alone (50% total PA biofilm reductions) (Figure 5). Overall, phage-TOB combinations, were 27.7% more effective in reducing PA biofilm biomasses than TOB alone. The most effective phage was phage Φ14_OBG combined with TOB, which reduced 50% of the PA biofilm biomasses compared with TOB alone (Figure 5).

### 3.7. Supernatant CF PA Phenotypes and Antibiotic Sensitivity Tests

In 24-h-old PA biofilms from 25 non-mucoid CF PA isolates treated with phages at MOI 1 and 100, mucoid phenotypes appeared in the biofilm supernatants (Figure 6). After 48-h-incubation at 37 °C, mucoid colonies were detected from 30 phage-CF PA combinations (Pa_Ph1 vs. all five phages, Pa_Ph3 vs. Φ9_ZP2 and Φ17_OBG, Pa_Ph6 vs. Φ9_ZP2 and Φ14_OBG, Pa_Ph13 vs. Φ4_ZP1, Φ14_OBG and Φ17_OBG, Pa_Ph14 vs. Φ4_ZP1, Φ9_ZP2 and Φ17_OBG, Pa_Ph15 vs. all five phages, Pa_Ph18 vs. Φ17_OBG and Φ19_OBG, Pa_Ph23 vs. Φ9_ZP2, Φ14_OBG and Φ17_OBG, Pa_Ph33 vs. all five phages). After 3 passages on MacConkey agar plates, we observed stable CF PA phenotypes except for five combinations Pa_Ph3 vs. Φ17_OBG, Pa_Ph6 vs. Φ9_ZP2 and Φ14_OBG, and Pa_Ph14 vs. Φ4_ZP1 and vs. Φ17_OBG. Even though CF Pa_Ph3 exhibited the mucoid phenotype only when treated with Φ9_ZP2 at MOI 1 and with Φ17_OBG at MOI 100, the other eight CF PA isolates showed a mucoid phenotype unrelated to the phage or the MOI tested. The CF PA mucoid phenotypes were unrelated to the increase in biofilm biomass. Experiments searching the CF PA mucoid colonies in the biofilm supernatants also from the phage-untreated PA biofilms, failed to detect a switch towards a CF PA mucoid phenotype.

The antibiotic susceptibility test investigating eight mucoid CF PA variants showed that ancestral (wild type, WT) CF PA non-mucoid isolates and phage-induced (mucoid) CF PA variants had similar profiles with no significant trend. Depending on the PA isolates, those that switched towards the PA mucoid phenotypes lost or acquired resistance to selected antibiotics (Table 3).

### 3.8. Phage-CF PA Planktonic Cell Infection Kinetics

Using phages Φ4_ZP1 and Φ14_OBG at MOI 10 to treat Pa_Ph4 planktonic cell cultures, and Φ4_ZP1 and Φ9_ZP2 at MOI 10 to treat Pa_Ph6 planktonic cell cultures in in vitro experiments to assess whether phage exposures increased PA biofilm biomasses owing to bacterial regrowth, phage lytic activity reached its highest level 6–7 h after exposure, and bacterial regrowth followed (Figure 7). At 24 h after phage exposures, treated and untreated PA controls exhibited identical bacterial regrowth levels. During the same observation period, the phage concentration increased by about 2-log, from ~10^8^ to ~10^10^ PFU/mL.

After Pa_Ph4 was exposed to phages Φ4_ZP1 and Φ14_OBG for 6 h, a new PA morphotype emerged characterized by a “smaller and rounder” shape than the ancestral type (Figure 7). After only one passage on MacConkey agar plate without phage selective pressure, this new PA phenotype reverted to the ancestral type.

Antibiotic activity susceptible at a standard dosing regimen (bold blue numbers); susceptible at an increased exposure (green numbers); resistant (black numbers) (according to the European Committee on Antimicrobial Susceptibility Testing (EUCAST) guidelines (EUCAST, Version 11.0, 2021, https://www.eucast.org/clinical_breakpoints/, accessed on 15 January 2021).

PA isolates tested for antibiotic susceptibility: Pa Ph1, Pa_Ph3, Pa_Ph13; Pa_Ph14, Pa_Ph15, Pa_Ph18, Pa_Ph23, and Pa_Ph33. Phages inducing MUC Pa phenotypes: Φ4_ZP1, Φ4; Φ9_ZP2, Φ9; Φ14_OBG, Φ14; Φ17_OBG, Φ17; Φ19_OBG, Φ19. The numbers indicate diameters in millimeters observed by the Kirby–Bauer diffusion test, reported in colors according to EUCAST 2021. Antibiotics tested: Amikacin, AMK; aztreonam, ATM; cefepime, FEP; ceftazidime, CAZ; ciprofloxacin, CIP; imipenem, IPM; levofloxacin, LVX; meropenem, MPM; piperacillin-tazobactam, TZP; and tobramycin, TOB.

## 4. Discussion

All 22 lytic bacteriophages newly isolated from five environmental sewage sources we tested against the 32 clinical CF PA isolated at various lung infection stages from the sputum of patients with CF successfully lyse up to 93.7% of CF PA isolates (Appendix A). By testing lytic phage activity in vitro, we identified five lytic phages all showing wide host range spectra against the panel of 32 CF PA isolates, namely Φ4_ZP1, Φ9_ZP2, Φ14_OBG, Φ17_OBG, and Φ19_OBG (Figure 2). When we tested in vitro the CF PA biofilms with three antibiotics chosen for their efficiency against CF PA isolates [8,16], after exposing them to phages for 4 h, only one of the five phages selected, the phage Φ4_ZP1, combined with MPM and compared with MPM alone, exhibited major lytic activity in significantly reducing two clinical CF PA biofilms (Pa_Ph10 at MOI 100, *p* < 0.01; Pa_Ph12 at MOI 1, *p* < 0.01) (Figure 4). This key unexpected finding contrasts strongly with previous evidence in phage-MPM combinations reporting no improved antimicrobial activity against various non-CF PA biofilms [17]. Another phage-antibiotic combination that in our tests in vitro proved significantly more efficient at diminishing PA biofilm biomasses than TOB alone, is the combination phage Φ14_OBG-TOB against Pa_Ph3 and Pa_Ph7 at MOI 100 (*p* < 0.001) (Figure 5). These new findings on our successful phage-antibiotic combinations, MPM and TOB, in in vitro experiments, contrast with a previous study reporting that a phage-TOB combination failed to reduce the PAO1 biofilm [18]. Our study now brings this result into question by providing in vitro evidence that Φ4_ZP1-MPM and Φ14-TOB combinations compared with antibiotics alone against clinical CF PA isolates from various lung infection stages exhibit an improved effect in reducing PA biofilm biomasses (Figure 4 and Figure 5). Even though MPM and TOB combined with two of the five selected phages effectively reduced PA biofilms, Φ17_OBG-AMK combination was less efficient than AMK alone in reducing two clinical CF PA biofilms (Figure 3, Figure 4 and Figure 5). Nevertheless, to test whether phages combined with antibiotics exerted improved effects, we chose the three major antibiotics able to treat CF PA biofilms efficiently. This choice, unlike a phage-antibiotic combination that proved unable to lyse a laboratory PA biofilm [17,18], provides new evidence on identical increased activity of phage-MPM and phage-TOB combinations in reducing CF PA biofilms. This new finding should be useful to advance evidence on in vitro reduction in phage resistance and PA bacterial populations in laboratory PA. We suggest that newly isolated environmental phages should be identified from several sewage sources, including hospital sewage, and combined with MPM and TOB to test several CF PA biofilms [16,18,19,20].

For proper phage selection, another important finding in our in vitro study is that the phage-antibiotic improved effect is phage MOI unrelated, but phage-CF PA isolate, and phage-antibiotic related [20]. Advancing earlier findings on phage-antibiotic combinations against PA biofilms [8], future in vitro CF PA experiments need to amplify patient-tailored approaches. Hence, when testing newly isolated phages, preferably coming from sewage sources in hospitals where patients with CF are treated, we should aim to identify the most efficient phage and the right antibiotic timing, thus ensuring the best phage-antibiotic combinations [10]. This key concern highlighted from our in vitro results on the Φ4_ZP1-MPM combination underlines the significant CF PA biofilm dispersion owing to the combined phage-antibiotic lytic activity. Hence, to avoid futile tests on laboratory PA isolates, we favor in vitro tests on phages combined with MPM, and other carbapenem antibiotics against several clinical CF PA biofilms.

What is especially important in advancing information on PA phenotype switching from non-mucoid to mucoid morphotypes in the lungs of patients with CF, is that our in vitro tests, exposing nine CF non-mucoid PA biofilms to the five selected newly isolated environmental phages, produce an astonishing 30 new mucoid PA colonial morphotypes emerging in the CF PA phage-infected biofilm supernatants. These new mucoid PA morphotypes that switched from their matched PA wild types seem mostly unrelated to the single phage exposures. Even though no evidence yet proves whether new mucoid PA morphotypes originate from a possible phage-imposed selection in vivo [21], this key finding from our in vitro experiments on CF PA isolates supports ample previous observations on new mucoid CF PA morphotypes emerging in chronic PA lung infections [1,4] possibly directly related to phage selection in vivo. To advance the field on testing phages in future CF PA in vivo, further in vitro experiments should therefore investigate phages against CF PA biofilms from early and long-term infections, rather than testing in vain phages only against laboratory PA. Doing so might risk hiding possibly dangerous phage-induced side effects in switching PA wild-types to PA mucoid morphotypes, and counteracting potential in vivo adjunctive therapeutic phage strategies [21,22]. In our experiments, this dangerous event arose 6 h after we exposed Pa_Ph4 to phages Φ4_ZP1 and Φ14_OBG, and exhibited a PA morphotype characterized by a “smaller and rounder” shape than the ancestral type. After only one passage on MacConkey agar plate, without phage selective pressure, this new PA phenotype reverted to the ancestral type, implying that it is not a small-colony-variant CF PA.

When we tested in vitro the newly isolated lytic phages against CF PA biofilms and CF PA planktonic cell cultures, several key undesirable events appeared. These were CF PA biofilm biomass overproduction, and a fast bacterial regrowth that began within 24 h after we exposed CF PA to phages. We deem this in vitro observation a crucial concern that reinforces past evidence on the newly emerging CF PA phenotypes, and the astonishing PA defense strategies against phages at each CF lung infection stage [23]. Notwithstanding phage exposures, this undesirable finding explains past evidence on the natural PA plasticity that enables switching to new phenotypes, hence provoking PA biofilm overproduction, and repopulating PA biofilms and cultures in laboratory PA, and only one CF PA biofilm [12,24,25]. Other studies on non-CF MDR PA and few CF PA biofilms reported contrasting successful results on reducing PA biofilm biomass, and avoiding fast bacterial regrowth [8,26,27,28]. Conversely, among the major research advances our tests on phage lytic efficacy in treating 24-h-old CF PA biofilms offer, are the numerous CF PA strains tested with phages, thus supporting past evidence on the unwelcome increase in CF PA biofilms and fast bacterial regrowth [12,24,25]. Owing to these contrasting results, to advance new in vitro experiments, we strongly suggest methodologically well-designed studies testing phage lytic efficacy in vitro and in vivo against several CF PA biofilms including various strategic approaches or phage-PA exposures by testing phages isolated as they were in our experiments, or trained phages in cocktails [10].

Of extreme importance in any study on lytic phage activity against CF PA isolates and biofilms, is to select newly isolated phages from several environmental sewage sources including hospital sewage and to test phages against chronic, rather than early, CF PA. Counteracting previously in vitro successful findings on lytic phage activity against CF PA isolates [29,30], even though we tested 22 newly isolated environmental phages against several CF PA isolates, when we tested phages alone against chronic CF PA in vitro, phages showed generally poor PA lysis. We explain this disappointing finding by our choice in using numerous newly isolated environmental phages against several PA isolates from chronically PA colonized patients with CF [29,30]. Over time, in these patients, PA undergoes changes and adapts to the worsening lung environment by developing new mucoid phenotypic strains, thus becoming progressively more resistant to phages than do PA in early CF infections [31]. From these important novel in vitro findings, we deem that future experiments on phages against CF PA should use a two-step approach: First, phages should be isolated from hospital sewage, and identified on chronic clinical PA isolates, such as IB; second, specific phages should be tested combined with MPM or other carbapenems tailored on patient specific CF PA isolates and MIC, so as to target the right phage for the right patient with CF (personalized phage-therapy) [10].

Another major finding obtained by isolating phages and testing them against several CF PA isolates, is the enhanced cross-activity against IB, laboratory, and clinical PA. This promising result provides new insights on possible phage adaptation “training” able to enhance cross-activity against specific patient CF PA strains, also envisaging personalized phage cocktails [32]. In agreement with these intriguing experiments [8,10], we agree with researchers suggesting that the most efficacious lytic phages are those isolated on purpose, personalized, against specific antibiotic-resistant PA strains in patients with CF (*sur-mesure*, patient-tailored therapy) [33,34,35].

Our in vitro study has limitations. First, we tested phage lytic efficacy on a static CF PA biofilm model rather than a dynamic model [10]. To bridge the current information gap on phage experiments in CF PA, we deem it more appropriate to undertake in vitro phage experiments on a static CF PA biofilm model, like that previously successfully used in in vitro laboratory PA and non-CF PA [10]. Second, our in vitro study, although we successfully isolated 22 new environmental phages, showed wide lytic activity in relatively few. Despite this deficiency, owing to their lytic activity against numerous (32) CF PA isolates, we stored all the 22 newly isolated environmental phages in our bank at OBG in Rome, Italy, for further investigations. These investigations will include training and mixing phages in cocktails for personalized formulations to test in vitro and in vivo phages alone, or combined with carbapenem antibiotics, against several other CF PA isolates and biofilms.

## 5. Conclusions

Our in vitro experiments provide new information on newly isolated environmental phages from various sewage sources, including one from our hospital treating patients with CF. To bridge the gap for supporting the use of phages alone or combined with antibiotics (MPM and TOB) against CF PA infections in in vitro and further in vivo experiments, our in vitro tests raise concerns on investigating phage treatments against CF PA alone, because phages alone appear unable to replace antibiotics in treating CF PA isolates and PA biofilms. Given that our results suggest that phage treatment could increase rather than decrease CF PA biofilm biomasses, and PA might switch towards an untreatable, dangerous PA mucoid variant phenotype, future in vitro and in vivo tests to improve phage lytic efficacy in CF should include phages identified mainly from hospital sewage sources, trained and mixed in cocktails, and combined with antibiotics (carbapenems) against several CF PA isolated from patients at various lung infection stages.

## Figures and Tables

**Figure 1 microorganisms-09-00478-f001:**
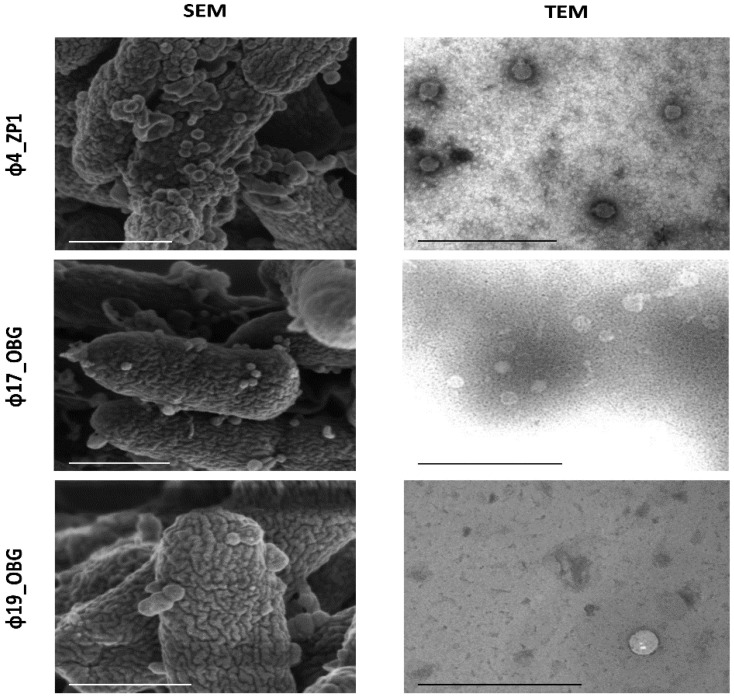
Scanning electron microscopy (SEM) and transmission electron microscopy (TEM) phage characterization. Phages Φ4_ZP1, Φ17_OBG, and Φ19_OBG exhibited icosahedral heads and had tails. Scale-bars = 500 nm.

**Figure 2 microorganisms-09-00478-f002:**
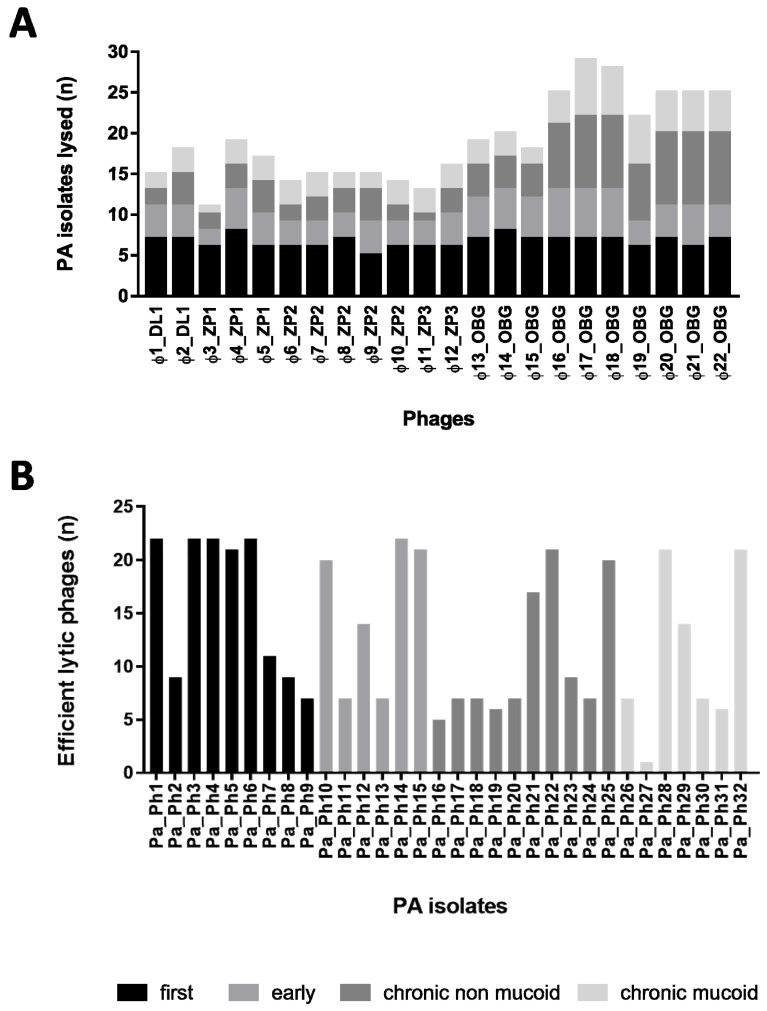
Phage cross activities (lytic abilities) of the 22 newly isolated phages against 32 *Pseudomonas aeruginosa* (PA) isolates (Pa_Ph1–Pa_Ph32) from patients with cystic fibrosis (CF); (*n*) = numbers. Phages are stratified in (**A**), and CF PA isolates in (**B**) according to the various lung infection stages: Black for first infection (i.e., CF PA isolated right after the first infection); light grey, early infection (i.e., early CF PA isolated one year after the first infection); grey, chronic non-mucoid CF PA isolates, and very light grey, chronic mucoid CF PA isolates (i.e., chronic CF PA isolated for at least five years after first infection). Overall, whereas all the newly isolated environmental phages showed the same lytic activity against several first CF PA infections, phages (Φ16_-Φ22_OBG) isolated from the Bambino Gesù (OBG) sewage source showed lytic ability against 20 or more CF PA isolates tested, and the best lytic activity against chronic non-mucoid or mucoid CF PA isolates (**A**,**B**).

**Figure 3 microorganisms-09-00478-f003:**
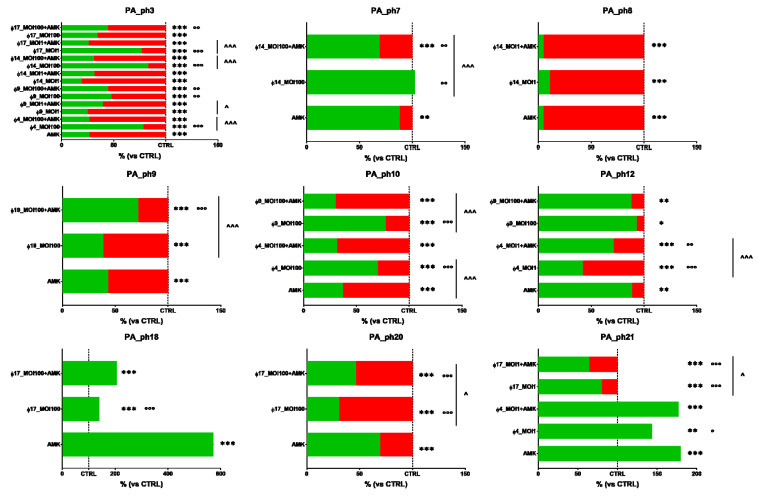
In vitro phage and amikacin (AMK) activity, alone and in combination, against nine 24-h-old cystic fibrosis *Pseudomonas aeruginosa* (PA) biofilms. Biofilm dispersion evaluated by crystal violet stain. The results are shown as percentages in dispersed PA biofilms (highlighted in red) compared with unexposed PA control samples (CTRL) in trypticase soy broth (TSB); the dotted line indicates 100% residual biofilm after a challenge with TSB; significant levels * *p* < 0.05, ** *p* < 0.01, *** *p* < 0.001 each treatment vs. CTRL; ° *p* < 0.05, °° *p* < 0.01, °°° *p* < 0.001 each treatment vs. AMK, and ^ *p* < 0.05, ^^^ *p* < 0.001 each treatment vs. phage-AMK analyzed by the χ^2^ test.

**Figure 4 microorganisms-09-00478-f004:**
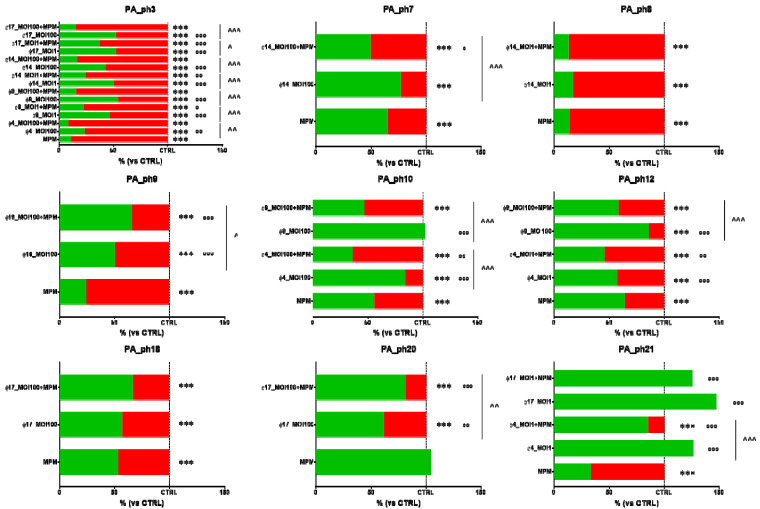
In vitro phage and meropenem (MPM) activity, alone and in combination, against nine 24-h-old cystic fibrosis *Pseudomonas aeruginosa* (PA) biofilms. Biofilm dispersion evaluated by crystal violet stain. The results are shown as percentages in dispersed PA biofilms (highlighted in red) compared with unexposed PA control samples (CTRL) in trypticase soy broth (TSB); the dotted line indicates 100% residual biofilm after a challenge with TSB; significant levels *** *p* < 0.001 each treatment vs. CTRL; ° *p* < 0.05, °° *p* < 0.01, °°° *p* < 0.001 each treatment vs. MPM, and ^ *p* < 0.05, ^^ *p* < 0.01, ^^^ *p* < 0.001 each treatment vs. phage-MPM analyzed by the χ^2^ test.

**Figure 5 microorganisms-09-00478-f005:**
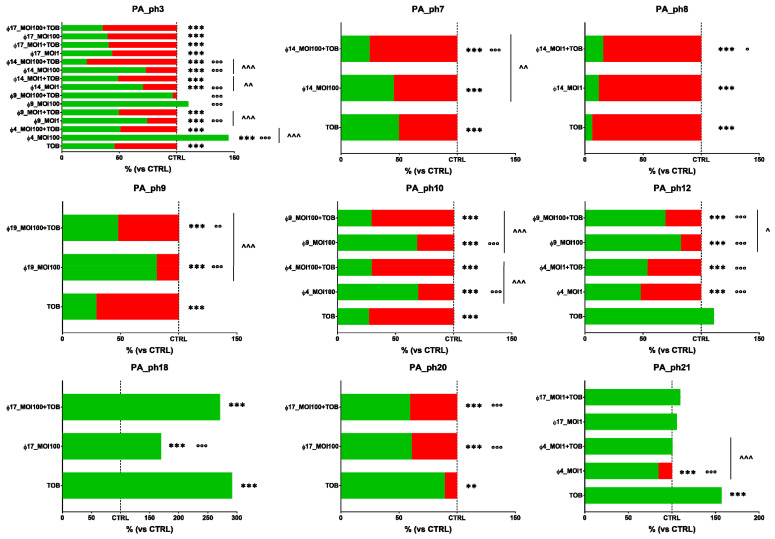
In vitro phage and tobramycin (TOB) activity, alone and in combination, against nine 24-h-old cystic fibrosis *Pseudomonas aeruginosa* (PA) biofilms. Biofilm dispersion evaluated by crystal violet stain. The results are shown as percentages in dispersed PA biofilms (highlighted in red) compared with unexposed PA control samples (CTRL) in trypticase soy broth (TSB); the dotted line indicates 100% residual biofilm after a challenge with TSB; significant levels ** *p* < 0.01, *** *p* < 0.001 each treatment vs. CTRL; ° *p* < 0.05, °° *p* < 0.01, °°° *p* < 0.001 each treatment vs. TOB, and ^ *p* < 0.05, ^^ *p* < 0.01, ^^^ *p* < 0.001 each treatment vs. phage + TOB analyzed by the χ^2^ test.

**Figure 6 microorganisms-09-00478-f006:**
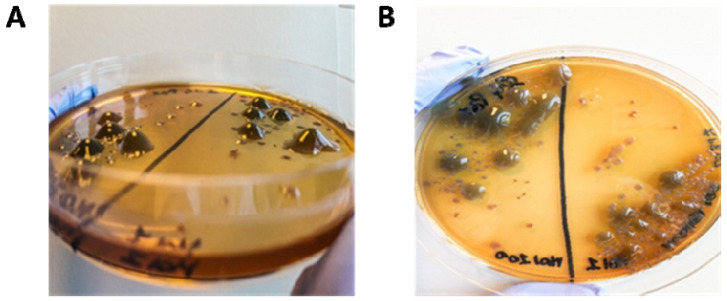
Mucoid *Pseudomonas aeruginosa* (PA) phenotypes observed in the biofilm supernatants exposed to phages. (**A**,**B**) Blob-like, raised mucoid colonies, after 48-h-incubation at 37 °C, grown on MacConkey agar after exposing preformed PA biofilm to phages for 24 h.

**Figure 7 microorganisms-09-00478-f007:**
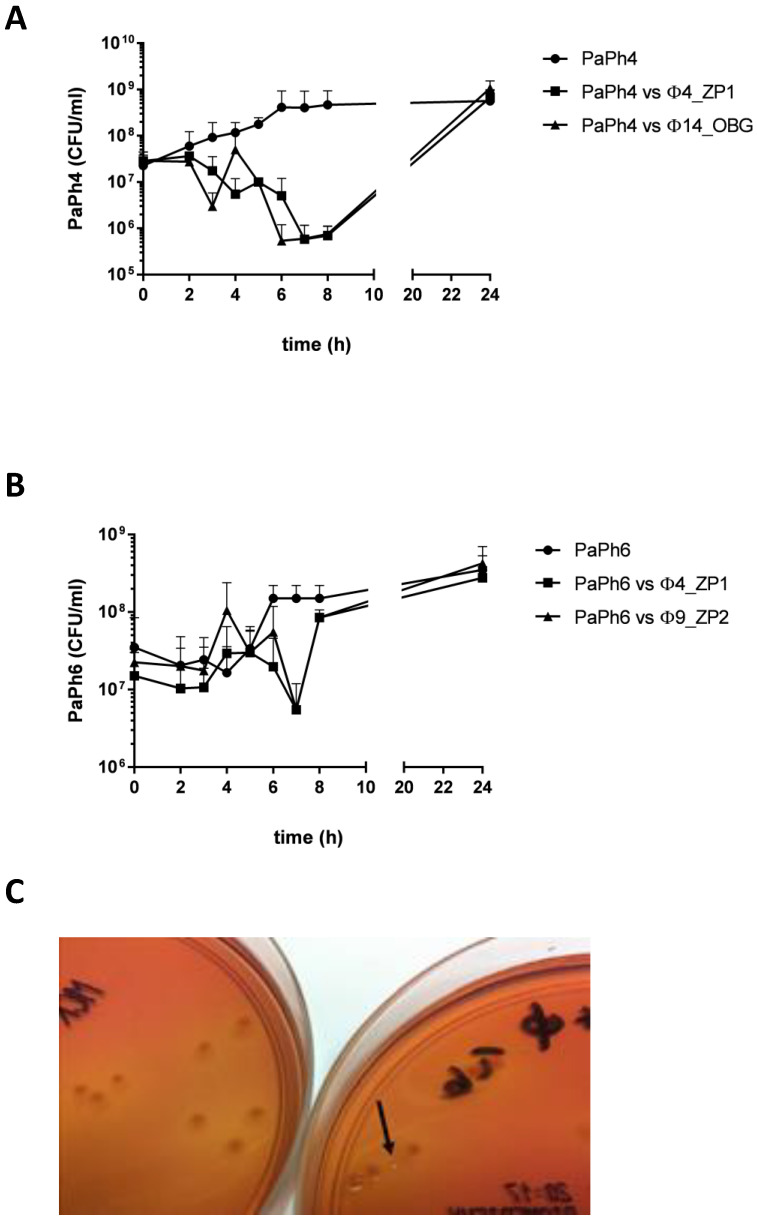
Phage in vitro activity against planktonic *Pseudomonas aeruginosa* (PA) cells. (**A**), the Pa_Ph4 isolate exposed to phages Φ4_ZP1 and Φ14_OBG at MOI 10. As control, Pa_Ph4 grown in medium alone. (**B**), the Pa_Ph6 isolate exposed to phages Φ4_ZP1 and Φ9_ZP2 at MOI 10. As control, Pa_Ph6 grown in medium alone. (**C**), the “small round” PA morphotype observed on MacConkey agar plates after exposing Pa_Ph4 to the phage Φ4_ZP1 for 6 h (arrow on the right plate). Pa_Ph4 after 6-h culture without the phage (on the left plate).

**Table 1 microorganisms-09-00478-t001:** Phage morphology at transmission electron microscopy and scanning transmission electron microscopy image analyses and growth characterization for the five newly isolated environmental lytic phages selected.

Phage Names	Phage Morphologic and Growth Characteristics				
	Head Length (nm) ^a^	Tail Length (nm) ^a^	Burst Size ^b^ (PFU/Cell Mean ± SD)	Latent Period (min)	Infection Cycle (min)
Φ4_ZP1	75	22	8.5 ± 3.0	10	20
Φ9_ZP2	64	20	11.0 ± 3.0	30	45
Φ14_OBG	72	35	12.7 ± 3.7	35	45
Φ17_OBG	64	20	34.0 ± 18.0	20	50
Φ19_OBG	89	21.5	4.0 ± 0.5	40	45

^a^ Median values in head and tail measurements are expressed in nanometers, nm, after at least six measurements. ^b^ Burst sizes are expressed as plaque forming unit (PFU)/cell mean ± standard deviation (SD) after three independent experiments.

**Table 2 microorganisms-09-00478-t002:** In vitro activity for the five newly isolated lytic phages (Φ4_ZP1, Φ9_ZP2, Φ14_OBG, Φ17_OBG, Φ19_OBG) at multiplicity of infection (MOI) 1, and MOI 100 tested against the 24-h-old-preformed *Pseudomonas aeruginosa* (PA) biofilms.

Significant PA Biofilm Reduction Numbers (%)				
	4 h		24 h	
Phages Tested	MOI 1	MOI 100	MOI 1	MOI 100 ^a^
Φ4_ZP1	7/18 (38.8)	9/18 (50.0)	10/21 (47.6)	7/21 (33.3) ^a^
Φ9_ZP2	4/15 (26.6) ^b^	10/15 (66.6) ^b^	6/17 (35.2)	8/17 (47.0)
Φ14_OBG	6/18 (33.3)	10/19 (52.6)	10/22 (45.4)	15/22 (68.1) ^a,c^
Φ17_OBG	14/25 (56.0)	12/26 (46.1)	10/31 (32.2)	10/31 (32.2) ^c^
Φ19_OBG	11/24 (45.8)	14/24 (58.3)	6/24 (25.0)	10/24 (41.6)
Total	42/100 (42.0)	55/102 (53.9)	42/115 (36.5)	50/115 (43.4)

PA biofilms from PAO1, and 33 cystic fibrosis (CF) Pa_Ph isolates after 4 and 24 h exposures. Numbers and percentages (%) of statistically significant PA biofilm reductions in crystal-violet stained samples. Statistically significant difference by χ^2^ test: ^a^ Φ4_ZP1 vs. Φ14_OBG *p* < 0.05; ^b^ Φ9_ZP2: MOI 1 vs. MOI 100 *p* < 0.001; ^c^ Φ14_OBG vs. Φ17_OBG *p* < 0.01.

**Table 3 microorganisms-09-00478-t003:** In vitro antibiotic activity comparing ancestral cystic fibrosis (CF) *Pseudomonas aeruginosa* (Pa) isolates (wild type, WT), and their switched mucoid (MUC) Pa phenotypes induced by phage exposures.

	Pa Isolates from Patients with CF and the Five Phages (Φ) Inducing MUC Pa Phenotype Switching															
	Pa_Ph1		Pa_Ph3		Pa_Ph13		Pa_Ph14		Pa_Ph15		Pa_Ph18		Pa_Ph23		Pa_Ph33	
Antibiotics	WT	MUC (Φ19)	WT	MUC (Φ9)	WT	MUC (Φ17)	WT	MUC (Φ4)	WT	MUC (Φ19)	WT	MUC (Φ19)	WT	MUC (Φ14)	WT	MUC (Φ19)
**AMK**	*19*	*25*	*24*	*25*	*24*	*21*	*21*	*21*	*22*	*19*	*17*	*20*	*19*	*22*	*24*	*20*
**ATM**	32	25	22	23	26	20	23	**0**	20	24	**16**	24	19	22	26	22
**FEP**	**17**	26	25	30	26	**20**	26	**0**	22	**19**	**11**	21	**17**	**19**	**15**	**18**
**CAZ**	26	28	24	28	28	20	24	**0**	24	20	18	24	18	20	28	**16**
**CIP**	**24**	28	35	28	30	**15**	28	28	**0**	29	**13**	26	**24**	**16**	30	**25**
**IPM**	32	24	23	27	21	**14**	*23*	**7**	25	26	**0**	26	**0**	**12**	21	25
**LVX**	**18**	23	24	24	24	**13**	24	**19**	**0**	**18**	**0**	**20**	**20**	**10**	24	**19**
**MPM**	*40*	*24*	*29*	22	*34*	**12**	*28*	**0**	*25*	*28*	**16**	*35*	**0**	**12**	*28*	*30*
**TZP**	35	27	24	28	28	18	25	**0**	28	**16**	24	**16**	19	**16**	26	20
**TOB**	**17**	*30*	*23*	*27*	*23*	*24*	*21*	*23*	**0**	*21*	*21*	*23*	*21*	*26*	*24*	*23*

PA isolates tested for antibiotic susceptibility: Pa Ph1, Pa_Ph3, Pa_Ph13; Pa_Ph14, Pa_Ph15, Pa_Ph18, Pa_Ph23, and Pa_Ph33. Phages inducing MUC Pa phenotypes: Φ4_ZP1, Φ4; Φ9_ZP2, Φ9; Φ14_OBG, Φ14; Φ17_OBG, Φ17; Φ19_OBG, Φ19. Antibiotics tested: Amikacin, AMK; aztreonam, ATM; cefepime, FEP; ceftazidime, CAZ; ciprofloxacin, CIP; imipenem, IPM; levofloxacin, LVX; meropenem, MPM; piperacillin-tazobactam, TZP; and tobramycin, TOB. The numbers indicate diameters in millimeters of the inhibition zone observed by the Kirby–Bauer diffusion test. Antibiotic activity susceptible at a standard dosing regimen (*italics*); susceptible at an increased exposure (underlined); resistant (**bold**) according to “The European Committee on Antimicrobial Susceptibility Testing (EUCAST). Breakpoint tables for interpretation of MICs and zone diameters. Version 11.0, 2021, https://www.eucast.org)”.

## Data Availability

The data presented in this study and in the Appendix A are available on request from the corresponding author.

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
