# Peer review of "In Vitro Newly Isolated Environmental Phage Activity against Biofilms Preformed by *Pseudomonas aeruginosa* from Patients with Cystic Fibrosis"

_microorganisms, 2021, doi:10.3390/microorganisms9030478_

Round 1

Reviewer 1 Report

This paper describes extensive studies on the identification and function of novel bacteriophages that infect Pseudomonas aeruginosa strains isolated from cystic fibrosis patients. The paper also reports on the combination of antibiotics with phages to kill the bacteria.

The paper can become acceptable for publication after some important revisions.

The burst size ratio should be written on a separate line as a formula

It is impossible to read the Figure 2 as it stands. The writing on each of the axes should be legible; in essence the figure should be enlarged.

Figures 3, 4 and 5 should also be made legible by splitting them and enlarging them.

Please describe how the data are interpreted to show a synergistic effect in some cases.

Long sentences that fill 4 lines should be avoided.

The Discussion section should be shortened as much as feasible. Scientists do not have much time to read long discussions.

Minor points

new PA colonial morphotypes -> new PA colony morphotypes

and incubated it at 37˚C -> and was incubated at 37˚C

assessed by spectrophotometer -> assessed by spectrophotometry

synergic effect -> synergistic effect

new unpublished evidence -> new evidence [since you are publishing it here, it is not unpublished]

The Conclusion section should not start with “In conclusion” because the phrase is redundant.

Author Response

Dear Editor and Reviewers,

We have answered all the reviewers’ critiques and have highlighted the changes in bold print in the new manuscript.

Reviewer 1

Q The burst size ratio should be written on a separate line as a formula

A We have provided a phage burst size formula as requested (Methods page 7).

Q It is impossible to read the Figure 2 as it stands. The writing on each of the axes should be legible; in essence the figure should be enlarged.

A We have reformatted Figure 2 to make it legible.

Q. Figures 3, 4 and 5 should also be made legible by splitting them and enlarging them.

Please describe how the data are interpreted to show a synergistic effect in some cases.

A. We have also made Figures 3,4, and 5 more legible by splitting and enlarging them.

We thank the Reviewer for the comment concerning “synergistic effect”. For clarity, in the revised version we replaced “synergistic” or “synergy” with “improved” (discussion section). This is consistent with our experiments that compared directly phage and phage-antibiotic combination for anti-biofilm effect rather than (phage alone)+(antibiotic alone) vs phage+antibiotic combinations by the chi-square test.

Q. Long sentences that fill 4 lines should be avoided.

A. We have shortened long sentences filling 4 lines (page 5, page 21).

Q. The Discussion section should be shortened as much as feasible. Scientists do not have much time to read long discussions.

A. Our discussion may be long, and we have left it so because it gives lay readers and scientists previously unavailable essential information enabling them to understand better how phages alone or combined with antibiotics should be used in future in vitro and in vivo experiments to treat CF PA biofilms from chronic infections.

Minor points

new PA colonial morphotypes -> new PA colony morphotypes

A. We have corrected “colonial” with “colony” in page 4 as requested.

and incubated it at 37˚C -> and was incubated at 37˚C

A: We have deleted “it” on page 7.

assessed by spectrophotometer -> assessed by spectrophotometry

A. We have changed spectrophotometer with spectrophotometry on page 8 and in each figure legend.

synergic effect -> synergistic effect

A. We have changed “synergic” with “improved”.

new unpublished evidence -> new evidence [since you are publishing it here, it is not unpublished]

A. We have deleted unpublished.

The Conclusion section should not start with “In conclusion” because the phrase is redundant.

A. We have deleted “in conclusion”.

Reviewer 2 Report

Overall comments:

This is a very interesting article looking at phages that can target CF PA clinical isolates. This is an understudied area and the findings are very interesting.

Specific comments: 

Throughout manuscript there is a lack of consistency between use of cystic fibrosis and CF (it seems to alternate randomly). Please make consistent.

Trypticase soy broth (TSB) and Pseudomonas aeruginosa (PA)  are defined many times in the manuscript and repeatedly in figure legends. It only needs to be defined once. Please fix.

Bottom page 7: define IB

Top page 8: What wavelength was measured when biofilm was assessed by spectrophotometer?

Figure 2: The X and Y axis are labeled with really tiny font making it hard to read. Then the key is in large font but poor resolution making all of the words "fuzzy". This figure needs to be remade with proper X and Y sized fonts and with all labels in high resolution.  The current figure is unacceptable.

Table 2: The title for Table 2 is way to long. Fix the table with a smaller yet descriptive title and move most of the information in the current title to under the table with all of the other extra information.

Figure 3 is poorly made and low resolution. Even after zooming in on the PDF, the scales could not be read. This figure needs to be remade with larger fonts on the graphs and also in high resolution so that the readers can actually read what the graphs say.

Figure 4: same comment as previous figure. Figure provided is low resolution and needs to be remade in high resolution with larger sized font on the graphs to allow reader to read the graphs.

Figure 5: same comment as previous figures. Increase font size and remake as a high resolution figure.

Figure 6. Provide a clear and general title before going through the images.  Make left image (A) and right image (B), then walk the reader through both A and B.

Figure 6: "grew onto" should be "grown on"

Bottom of page 20: two separate sentences discussing Table 3 are in their own paragraphs. Paragraphs should be more than one sentence.  Combine them into one paragraph and then smooth them out so they are less choppy.

Table 3: shorten title on Table 3. It is very long. Move some of that info to under section under the table.

Supplementary figures S1-S4: same as above: increase font size and remake in high resolution.

Author Response

Dear Editor and Reviewers,

We have answered all the reviewers’ critiques and have highlighted the changes in bold print in the new manuscript.

Reviewer 2

Q. Throughout manuscript there is a lack of consistency between use of cystic fibrosis and CF (it seems to alternate randomly). Please make consistent.

A. According to standard editorial practice, in abstract, text, figure and table titles or legends we already consistently extended the abbreviation CF for “cystic fibrosis” at first mention, then always used the abbreviation.

Q. Trypticase soy broth (TSB) and Pseudomonas aeruginosa (PA) are defined many times in the manuscript and repeatedly in figure legends. It only needs to be defined once. Please fix.

A. According to standard editorial practice, as in the aforementioned reply, we already extended abbreviations at first mention in the text and in each figure.

Q. Bottom page 7: define IB

A. We have defined indicator bacteria IB on page 4.

Q. Top page 8: What wavelength was measured when biofilm was assessed by spectrophotometer?

A. We have specified as follows “…were assessed spectrophotometrically by measuring optical density at 492 nm (OD492).”

Q. Figure 2: The X and Y axis are labeled with really tiny font making it hard to read. Then the key is in large font but poor resolution making all of the words "fuzzy". This figure needs to be remade with proper X and Y sized fonts and with all labels in high resolution.  The current figure is unacceptable.

A. We have increased the size of the font used for labeling the X and Y axes in Figure 2 and used high resolution labels.

Q. Table 2: The title for Table 2 is way to long. Fix the table with a smaller yet descriptive title and move most of the information in the current title to under the table with all of the other extra information.

A. We have shortened the title for Table 2 and moved most of the information from the title to the legend.

Q. Figure 3 is poorly made and low resolution. Even after zooming in on the PDF, the scales could not be read. This figure needs to be remade with larger fonts on the graphs and also in high resolution so that the readers can actually read what the graphs say.

Figure 4: same comment as previous figure. Figure provided is low resolution and needs to be remade in high resolution with larger sized font on the graphs to allow reader to read the graphs.

Figure 5: same comment as previous figures. Increase font size and remake as a high resolution figure.

A. We have remade Figure 3, 4 and 5 with larger fonts and high resolution to improve readability.

Q. Figure 6. Provide a clear and general title before going through the images.  Make left image (A) and right image (B), then walk the reader through both A and B.

Figure 6: "grew onto" should be "grown on"

A. We have provided a title to Figure 6 and changed “grew onto” to “grown on”.

Q. Bottom of page 20: two separate sentences discussing Table 3 are in their own paragraphs. Paragraphs should be more than one sentence.  Combine them into one paragraph and then smooth them out so they are less choppy.

A. We have combined the two sentences on page 20.

Q. Table 3: shorten title on Table 3. It is very long. Move some of that info to under section under the table.

A. We have shortened the title in Table 3 and moved some information to the legend.

Q. Supplementary figures S1-S4: same as above: increase font size and remake in high resolution.

A. We have remade Figure S1-S4 with larger fonts and high resolution to improve readability.